# Selectively Answering Ambiguous Questions

**Jeremy R. Cole**[1] and **Michael JQ Zhang**[1,2] and **Daniel Gillick**[1]
**Julian Martin Eisenschlos**[1] and **Bhuwan Dhingra**[1,3] and **Jacob Eisenstein**[1]
1. Google DeepMind
2. University of Texas
3. Duke University

## Abstract

Trustworthy language models should abstain from answering questions when they do not know the answer. However, the answer to a question can be unknown for a variety of reasons. Prior research has focused on the case in which the question is clear and the answer is unambiguous but possibly unknown. But the answer to a question can also be unclear due to uncertainty of the questioner's intent or context. We investigate question answering from this perspective, focusing on answering a subset of questions with a high degree of accuracy, from a set of questions in which many are inherently ambiguous. In this setting, we find that the most reliable approach to decide when to abstain involves quantifying repetition within sampled model outputs, rather than the model's likelihood or self-verification as used in prior work. We find this to be the case across different types of uncertainty and model scales, and with or without instruction tuning. Our results suggest that sampling-based confidence scores help calibrate answers to relatively unambiguous questions, with more dramatic improvements on ambiguous questions.

## 1 Introduction

Any practical Question Answering (QA) system must be able to assess its own confidence so that it can (1) avoid making up incorrect, incomplete, or misleading answers, and (2) request clarification to resolve ambiguity arising from unclear questions or missing context. In recognition of this desirable property, recent work has addressed confidence estimation of Large Language Models (LLMs) by evaluating QA tasks in unambiguous settings (Jiang et al., 2021; Kadavath et al., 2022). In these experiments, questions are posed to the model with multiple-choice answers so the probability of each answer can be computed. Here we extend the study of confidence to practical scenarios, requiring free-form text answers to arbitrary questions which may be underspecified or ambiguous.

We describe a range of experiments to help disentangle uncertainty about the world from uncertainty about the question. While we focus on the question answering capabilities of large pretrained language models, these two forms of uncertainty can be more clearly distinguished in an idealized scenario in which a symbolic question-answering system first converts the question into a formal denotation (such as an SQL query or lambda expression), which is then applied to a knowledge base (e.g., Zelle and Mooney, 1996). The system may be uncertain about the meaning of the question — *denotational uncertainty* — if the question is underspecified because of assumed background knowledge or context. For any given denotation, the system may also be uncertain about the correct answer — *epistemic uncertainty* — if the knowledge base is incomplete[1]. In Figure 1, denotational uncertainty is shown in the upper fork and epistemic uncertainty is shown in the lower set of forks.

The most competitive systems today do not construct explicit denotations of questions. As a result, it can be difficult to clearly separate these two forms of uncertainty. Nonetheless, our general approach is to attempt to first approximately disambiguate the question (in natural language) and then selectively answer user questions with sufficiently high confidence. This allows the model to more effectively represent ambiguous user inputs. This scheme is represented in Figure 1. Further, we argue that repeated sampling within our disambiguate-then-answer framework provides reliable confidence estimates of the model.

We summarize our contributions as follows:

1. We reframe the discussion of model confidence with respect to denotational and epistemic uncertainty and address each in turn.

---

[1]Gruber et al. (2023) present an overview of uncertainty in Machine Learning, making a related distinction between *epistemic* and *aleatoric* uncertainty. We do not address aleatoric uncertainty in this work.

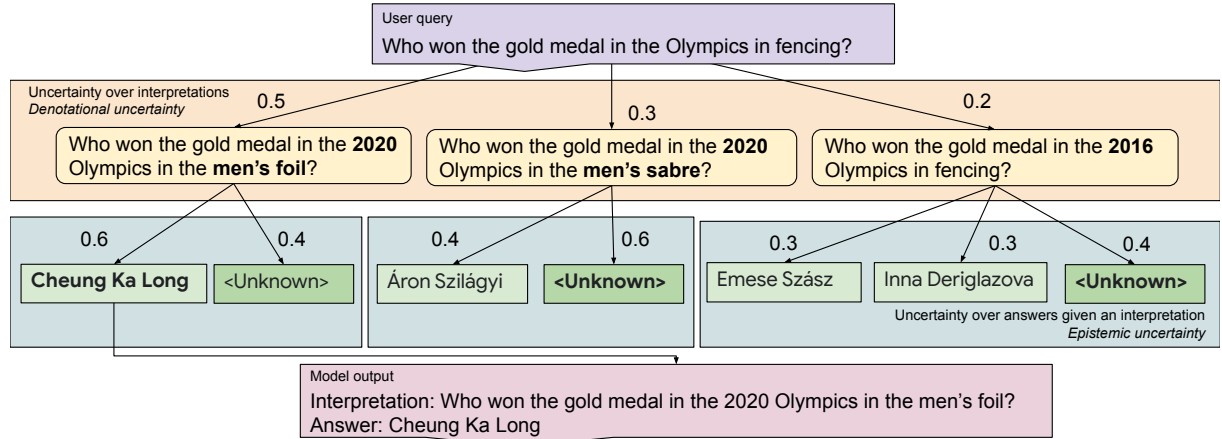

Figure 1: Uncertainty in Question Answering systems may arise in various ways. We propose a scheme called disambiguate then answer where the model first attempts to pose an unambiguous interpretation of the user question (yellow), then selectively produces an answer to this question, alternatively abstaining ("Unknown") (green). The log likelihood of the model is shown above each generation. In addition, we find that sampling multiple times from the model generally allows for more robust confidence estimates.

Our experiments suggest that the answer probabilities given by current LLMs are somewhat helpful for assessing epistemic uncertainty but cannot detect denotational uncertainty.

2. We present two simple and effective approaches for measuring confidence under all types of uncertainty, based on answer counts within repeated samples from the LLM. These approaches are better calibrated than the model's answer probabilities and a self-verification prompt, and are especially effective for ambiguous questions and for instruction-tuned model variants.

## 2 Calibration for Question Answering

A predictor is said to be well-calibrated if its predictions are accompanied by confidence scores and those scores are informative of the likelihood that the prediction is correct. Such confidence scores can be used to support applications such as selective prediction, in which the model can abstain from predicting when its confidence is low (Chow, 1957; El-Yaniv et al., 2010). In *selective question answering* (e.g., Kamath et al. 2020; Zhang et al. 2021b), we assign each question-answer pair $(q, a)$ a confidence score $s(q, a)$. The system's output is then parameterized by a threshold $\tau$,

$$\hat{y}_\tau(q) = \begin{cases} \arg\max_a s(q, a), & \max_a s(q, a) > \tau \\ \varnothing, & \text{else,} \end{cases}$$

(1)

with $\varnothing$ representing abstention. Given a probabilistic model $P(a \mid q)$, a natural choice for the confidence score $s(q, a)$ is the conditional probability of the answer. This and other scoring functions are discussed in Section 3.

For *information-seeking* queries such as those found in Natural Questions (Kwiatkowski et al., 2019), the askers have no answer in mind, which makes it more likely that the questions are accidentally ambiguous. The task of answering an ambiguous question requires solving at least two subtasks: determining the meaning of the question (its *denotation*), and then identifying the answer. This pipeline is shown in Figure 1; in a probabilistic model, this can be represented mathematically as $P(a \mid q) = \sum_d P(a \mid d) P(d \mid q)$, with $d$ indicating the denotation. While prior work has investigated the calibration of answer probabilities given by pretrained language models (e.g., Kadavath et al., 2022), it is not clear whether the relatively positive findings of these studies extend to questions with significant denotational ambiguity, as represented by datasets such as AmbigQA (Min et al., 2020) and SituatedQA (Zhang and Choi, 2021). We address this question in Section 6.4.

## 3 Confidence Scores

Before proposing confidence scores for question answering, it will be helpful to review how these answers are generated in practical settings. A language model over a finite vocabulary $\Sigma$ defines a probability distribution $X$ over sequences of to-

kens in $\Sigma^*$, by modeling the marginal probability of a sequence $p(\sigma_1, \cdots, \sigma_n)$ using the conditional probabilities of individual tokens via the chain rule $\prod_{i=1}^n p(\sigma_i | \sigma_{i-1}, \cdots, \sigma_n)$. For QA, answers are obtained by conditioning on the question $q$ and first sampling from $X$ (or a version of $X$ adjusted with modifications such as temperature) and then applying a normalization or extraction process to it in order to map it to an answer space $\mathcal{A}$. This function $f : \Sigma^* \to \mathcal{A}$ can be as simple as punctuation removal and lower-casing, or as complex as code execution or extracting the answer from a chain-of-thought (Wei et al., 2022).

Using this background, we can describe various methods for estimating the scores of predictions from a language model (Figure 2).

**Likelihood** The most basic method we can use is likelihood-based calibration by computing $p(\sigma | q) = \prod_{i=1}^n p(\sigma_i | \sigma_{i-1}, \cdots, \sigma_n, q)$ for a sequence sampled from $X$. This may be used to rank the answers from most confident to least confident based on the model's produced likelihood of that answer. However, in practical settings this method may not accurately reflect the probability of observing an answer for several reasons:

**(1)** Language models typically incorporate several inference-time biases to improve the quality of the outputs, such as nucleus sampling (Holtzman et al., 2020), top-k sampling (Fan et al., 2018), length penalties, length truncation, and temperature. These techniques affect the output distribution in ways that are hard or impossible to capture in the likelihood score (Zhang et al., 2021a).

**(2)** Even for decoders that do not incorporate inference-time biases, the likelihood function as defined by the auto-regressive decomposition might leak probability mass to infinite sequences (see Meister et al., 2023, Proposition 2.4). This implies that the model that we sample from might not be a valid probability distribution.

**(3)** Finally, the likelihood fails to account for the extraction step $f$, which requires an intractable marginalization step over the model outputs.

In view of these limitations, it is preferable to use sampling to estimate properties of the resulting marginal distribution. Formally, we can use an *Index of Qualitative Variation* (IQV) (Wilcox, 1973) to measure the statistical dispersion of discrete samples produced by the model. Practically, there are a many possible IQVs, so we have chosen two fairly different approaches for our experiments. In each case, we generate 10 sampled outputs (at a temperature of 0.5) and use exact match (after lowercasing and removing punctuation) for comparison among outputs. Naturally, there are many other ways to compare outputs, including token-level overlap or an answer-span equivalence model, but we leave these alternatives for future work. See also Kuhn et al. (2023) and Lin et al. (2023) for concurrent work investigating alternative IQVs.

**Sampling Repetition** Our first approach is based on Wilcox's classic *Variation Ratio* which measures deviation from the mode. Instead, we simply compute the fraction of times that the sampled outputs match the greedy output (Temperature $= 0.0$). The more samples match the greedy output, the more confident we are in the answer. We count the number of samples which *exactly* match the greedy answer. We experimented briefly with more sophisticated approaches based on the BERT based answer equivalence model from Bulian et al. (2022) as well as the summed F1 score across sampled answers and the greedy answers, but these approaches did not meaningfully improve the calibration.

**Sampling Diversity** Our second approach is based on the diversity of the samples, computed by $1 - \frac{\text{num\_unique}}{\text{num\_samples}}$. Here, our confidence is inversely proportional to the number of distinct samples and is estimated as zero if all samples are different. Note that while this works relatively well in practice, it is heavily dependent on the number of samples, as we do not expect the number of unique answers to scale linearly with the number of samples.

We note that some recent work (Meister et al., 2021; Arora et al., 2022) has also described estimating statistics of random variables induced by language models. This involved sampling without replacement and using importance weighting, but they seem to surpass Monte-Carlo estimates only with many samples and peaky distributions. We leave further exploration of these methods on our evaluations as well as other IQVs for future work.

**Self-Verification** Finally, we consider self-verification, in which the language model assesses its own confidence (Kadavath et al., 2022). This is a two-step process, where first the model is prompted to provide a list of possible answers given a question. Then, this list is fed back into the model,

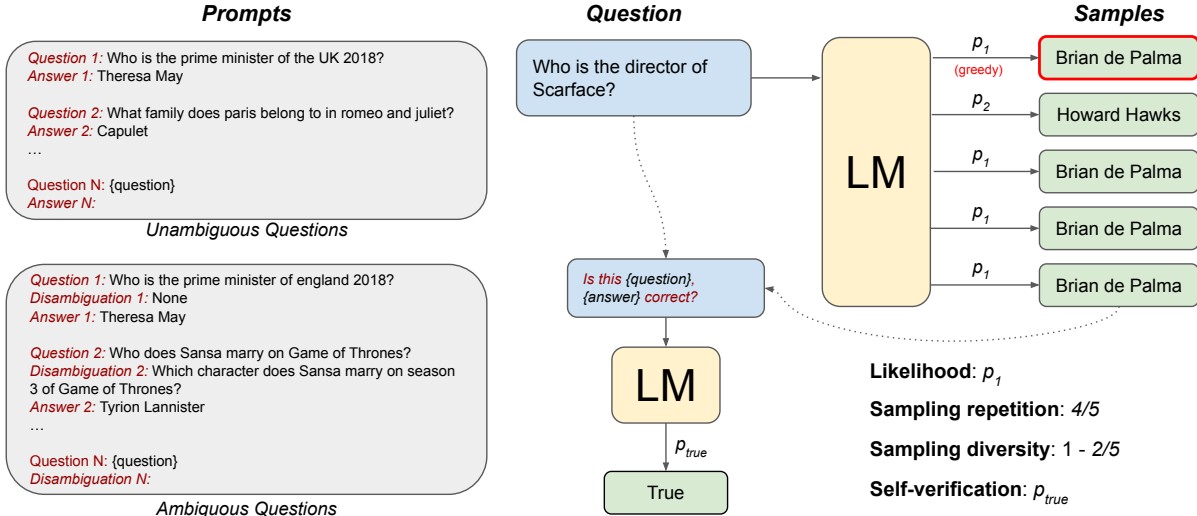

Figure 2: Methods for estimating the confidence of answers from an LM. *Sampling repetition* counts the number of times the greedy answer appears among samples from the LM. *Sampling diversity* counts the number of unique answers among samples from the LM. *Self-verification*, proposed by Kadavath et al. (2022), prompts the LM again with one of the sampled answers to measure the token-level probability of True. The prompts used for unambiguous and ambiguous questions are shown on the left—for the latter, we additionally prompt the model for disambiguations (omitted in the outputs shown on the right for brevity).

where the task is to determine if each answer is correct or incorrect. In practice, we re-use the sampled outputs from the previous run as the list of possible answers, and the greedy answer as the proposed answer. Then, the model scores the token "True", which we use to obtain a confidence score. In practice, these possible answers come from sampling with replacement, similar to the Sampling Repetition and Diversity methods, so the list of possible answers may contain duplicates. While these duplicates may affect the usefulness of the brainstorming exercise, the model is not choosing among them; instead, it is *only* scoring whether the greedy answer is correct, using the list of possible answers as context.

## 4 Evaluation setup

Our evaluations test the utility of the confidence scoring methods described in Section 3. We focus mainly on the pretrained language model PaLM, which obtains high accuracy on several question answering datasets using few-shot in-context learning (Chowdhery et al., 2022).

### 4.1 Metrics

There are many metrics for evaluating selective question answering systems, but fundamentally we want systems that (1) frequently return correct answers, and (2) rarely return incorrect answers.

These criteria are in conflict because any system can achieve zero error rate by always abstaining from answering. When the confidence score is a probability, we can also ask whether this probability is *well-calibrated*, in the sense that a confidence score of $s(q, a) = \alpha$ implies $\alpha$ is the probability of $a$ being the correct answer to $q$.

**Expected Calibration Error (ECE)** Predictions are grouped into ten equally sized bins, ranked by the evaluated system's assigned confidence scores. We compute the mean absolute distance between the average confidence score and the accuracy of predictions in each bin, averaging across all bins. If we interpret a confidence score to represent a probability, this corresponds to the difference in the predicted probability of correctness from the actual probability of correctness.

**ROC-AUC** Area under the receiver operating characteristic curve evaluates the uncertainty estimate's diagnostic ability as a binary classifier for correct and incorrect predictions by integrating over the tradeoff curve between the rates of true and false positives.

**Coverage@Acc** While ECE and ROC-AUC assess absolute and relative calibration respectively, we want a metric closely aligned with a practical use case: selective answering above a confidence threshold. To this end, we measure the fraction

of questions the system can answer correctly if it needs to maintain a certain accuracy. Specifically, C@ Acc is the maximum coverage such that the accuracy on the C% of most-confident predictions is at least Acc%. For example, if C@80 = 20, then the system achieves at least 80% accuracy on its 20% most-confident predictions (and lower than 80% accuracy on its $X$% most-confident predictions for any $X > 20$). With user-facing systems in mind, we set an accuracy of 80% (C@80).

# 5 Unambiguous Questions

To start, we explore the case of unambiguous questions with a single answer. In this scenario, all uncertainty should be of the *epistemic* form; the *denotation* of the questions should be clear. Previous work with this scenario has focused on the domain shift setting (Kamath et al., 2020) where the goal is to avoid answering questions from other datasets. Other work has looked at reading comprehension with relatively small models (Zhang et al., 2021b; Jiang et al., 2021). Open-domain question answering is substantially more difficult than reading comprehension, which likewise makes calibration more challenging.

## 5.1 Datasets

We explore this setting using two datasets: TriviaQA (Joshi et al., 2017) and NQ-Open (Kwiatkowski et al., 2019; Lee et al., 2019) because they are widely used for few-shot question answering. However, while it might be assumed that each question in these datasets has only one possible interpretation, this is often not the case. In fact, AmbigQA (Min et al., 2020) finds that over 50% of NQ-Open questions are underspecified. As such, we limit ourselves to unambiguous questions as annotated in AmbigQA. We will discuss the extended setting that includes underspecified queries in Section 6.

## 5.2 Experiment Setup

Our investigation focuses on few-shot in-context learning with large language models (LLMs). Our primary prompt is composed of four question and answer pairs from the training sets of Natural Questions and TriviaQA, respectively (see Figure 2). To reduce variance across experiments, the example QA pairs are selected randomly so that each input to the model has a different version of the prompt. We compute metrics over the entire evaluation set

for both datasets. We discuss investigation of other prompts in Appendix B.

## 5.3 Calibration Results

Results can be found in Table 1 and Figure 3 . First, we observe that the verification method used in prior work is inferior across nearly all calibration metrics. For unambiguous Natural Questions, depending on the metric used, we see that likelihood and sample repetition both work relatively well as calibration metrics. For TriviaQA, we see that the sample repetition method works notably better: this is potentially due to the presence of ambiguous questions in TriviaQA, which we do not attempt to remove. This hypothesis is strengthened by experiments in the next section that show that sampling methods tend to improve calibration more in the presence of more ambiguity.

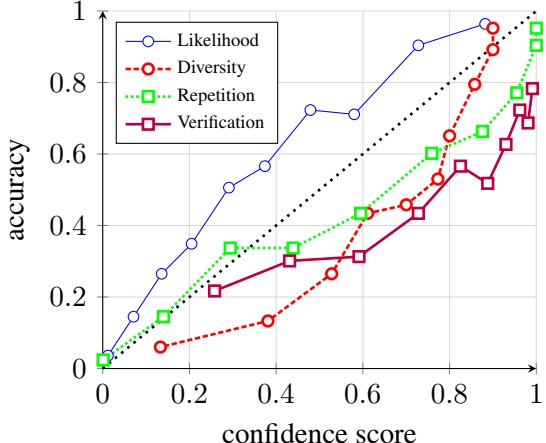

Figure 3: Plot of calibration error by comparing bucketed accuracy to bucketed confidence scores across methods. Plotted on the unambiguous portion of Natural Questions.

| Setting | M | EM | ROC-AUC | ECE | C@80 |
|---------|---|-----|---------|-------|------|
| NQ | L | | 0.843 | 0.141 | 44.2 |
| | D | 51.7 | 0.814 | 0.152 | 44.8 |
| | R | | 0.830 | 0.103 | **45.4** |
| | V | | 0.712 | 0.242 | 9.0 |
| TriviaQA | L | | 0.826 | 0.205 | 87.6 |
| | D | 72.5 | 0.829 | 0.080 | **88.1** |
| | R | | **0.844** | **0.052** | **88.1** |
| | V | | 0.740 | 0.199 | 81.8 |

Table 1: Non-Ambiguous QA Calibration on Natural Questions and TriviaQA. The column **M** refers to the calibration methods, abbreviated with (L)ikelihood, (D)iversity, (R)epetition, and (V)erification.

## 6 Ambiguous Questions

Questions can be ambiguous for a variety of reasons. Language is highly context-sensitive and meaning can be lost when context is missing. Often, interlocutors fail to clearly translate their intent into unambiguous language. Alternatively, ambiguity occurs due to a lack of alignment between interlocutors at a higher level of abstraction. Regardless of its origin, ambiguous questions give rise to denotational uncertainty, which we can conceptualize as a latent distribution over interpretations.

Realistic scenarios such as search and chat frequently contain ambiguous questions, which makes calibration more difficult because uncertainty may be epistemic, denotational, or both. Our approach will again involve sampling: this time over both interpretations and answers. Note that in this case, the approach has some similarity to self-consistency (Wang et al., 2022), as we compute sampling-based confidence scores over the final answers regardless of the exact interpretation.

### 6.1 Datasets

We evaluate using two datasets of underspecified queries that are derived from NQ-Open: AmbigQA and SituatedQA.

**AmbigQA** AmbigQA is created from a subset of Natural Questions (Kwiatkowski et al., 2019) that raters assess as ambiguous. After a rater deemed a question ambiguous, various unambiguous interpretations of the question and their respective answers are written. Thus, AmbigQA measures a model's ability to disambiguate-and-answer ambiguous questions. For instance, given a query like "Where does the new fallout game take place?", the model should be able to produce the disambiguated question "Where does the fallout 76 game take place?" and answer with "Appalachia". Models are evaluated based on the similarity between the generated question and the closest reference question as well as by producing the correct answer for the generated question.

**SituatedQA** SituatedQA is likewise created from a subset of existing questions and uses a relatively similar process, but focuses more narrowly on *temporal* and *geographic* ambiguity. As the type of ambiguity and the process used to create them is different, we generally evaluate them separately. The temporal questions are annotated with the time range that a question's answer is valid. Afterward,

additional time ranges and their corresponding answers are crowdsourced. The geographic questions are largely created by removing references to location and then crowdsourcing locations and corresponding answers. Note that SituatedQA does not provide metrics assessing the precision or possible recall of the disambiguations (though the recall would be particularly difficult to measure).

### 6.2 Experiment Setup

As above, we use prompts of question-answer pairs, with each example having six exemplars drawn from the training data. For AmbigQA, we use only the examples that have been further annotated by Stelmakh et al. (2022). For SituatedQA, we evaluate the Geographic and Temporal sets separately. For each example, three of the exemplars will be ambiguous; in those cases, the prompt will first include an interpretation of the question which is one of the provided disambiguations. In practice, we use the first disambiguation for each example.

### 6.3 Ambiguity Prediction

Before trying to assess calibration, we first seek to explore whether our method can predict whether a question is ambiguous according to AmbigQA. SituatedQA is not suitable for this task because its questions tend to follow templated patterns where the ambiguity arises from a missing time or place reference, while AmbigQA has more diverse expressions of ambiguity. We are not aware of prior work that evaluates ambiguity classification on AmbigQA or other datasets.

We use a first transformations of Section 3 to predict whether the model predicts the question is ambiguous. We discuss this in more detail in Appendix A. In short, none of the methods are particularly capable, with the best method achieving 58% accuracy when 53% of the questions are labeled as ambiguous.

Why is question ambiguity so difficult to predict? One possible explanation is that all queries are somewhat ambiguous. Queries are typically not *intentionally* underspecified: the questioner believes they have provided enough information to answer. In a conversation, this may fail and require developing common ground; in an information retrieval system, this may take the form of query refinement. All of the questions in the Natural Questions dataset were asked of the Google search engine and are answerable using Wikipedia

(Kwiatkowski et al., 2019), yet many are labeled as ambiguous by AmbigQA.

For instance, "Where is the world cup going to be?" might typically refer to the host country, but fans from the host country that year might instead refer to the cities the games will be played. Even binary choice questions like "Who won more games between the Packers and the Bears?" may appear unambiguous, but this question has different answers depending on the date it is asked.

In this sense, we propose that ambiguity is a function of both the interpreter and the query. However, measuring whether a model (or indeed, any interlocutor) understood a question is challenging. Often in dialogue, interlocutors may assume their counterpart understood if they receive a response similar to what they expected; alternatively, they may ask explicitly if they were understood.

This leads us to our disambiguate-then-answer paradigm: if there is no fundamental difference between ambiguous and unambiguous queries when taken out of the original context, then the model should first establish precisely what question it is attempting to answer.

### 6.4 Calibration Results

We evaluate calibration on ambiguous questions using the same methods described in Section 5 with results in Table 2 and Figure 4. We exclude the self-verification setup here, because its calibration is much worse on the unambiguous questions, and it is unclear how to translate the prompt when there are multiple interpretations of the question. Note that we consider the answer to be correct if it matches the answer of any of the disambiguations. We also present results for a "strict" matching setup, where we only consider the answer to be correct if it matches the answer of the closest disambiguation to the provided interpretation, as measured by token overlap.

Overall, the ambiguous questions are much more difficult than the unambiguous ones. In general, likelihood seems to become a worse method of measuring model uncertainty when the questions are ambiguous and sample repetition appears to improve calibration significantly.[2] The strict matching setup does not affect the ordering of the results, though numbers are lower overall.

---

[2]Note that the interpretations sometimes consist of longer sequences, thus confounding the usability of likelihood

## 7 Instuction Tuning and Scaling

Pretrained language models are typically fine-tuned before being used in applications. However, fine-tuning and other alignment techniques may disturb the model's calibration by emphasizing high performance on specific sets of tasks rather than the full pretraining distribution.[3] Table 4 shows that for Flan-PaLM (Chung et al., 2022), instruction tuning dramatically improves exact match accuracy while simultaneously worsening performance on confidence-based ranking (ROC-AUC) and calibration (ECE), when using model likelihood as the confidence score. However, this miscalibration can be mitigated by using sample-based confidence scores, which dramatically improves the calibration on ambiguous questions (AmbigQA). For the selective-prediction metric C@80, the instruction-tuned model with sampling is far ahead of any other method investigated. Note that we investigate Natural Questions and AmbigQA here as they are not in the instruction tuning training data.

We examine the effect of model scale on calibration, finding that in general, accuracy declines substantially in closed book question answering, but calibration stays roughly constant. See Appendix D for full results.

As sampling-based approaches require a linear increase in compute for the number of samples, we also examine how calibration scales with the number. In particular, we test three, five, and eight samples and compare that to the original results containing ten samples. Results can be found in Appendix E. Unsurprisingly, more samples seems to improve calibration, though it seems on the unambiguous Natural Questions slice, sampling diversity with a small number of samples works relatively well for the cost.

## 8 Related Work

This paper investigates calibration, selective question answering, and ambiguity within a single model. It builds on work across these topics. While AmbigQA (Min et al., 2020), ASQA (Stelmakh et al., 2022) and SituatedQA (Zhang and Choi, 2021) introduce new annotations for question answering datasets, neither these papers nor

---

[3]The GPT-4 technical report (https://cdn.openai.com/papers/gpt-4.pdf) shows that calibration is made significantly worse by the combination of instruction-tuning and reinforcement learning, but does not distinguish between these two factors.

| Dataset | M | Loose | | | | Strict | | | |
|---|---|---|---|---|---|---|---|---|---|
| | | EM | ROC-AUC | ECE | C@80 | EM | ROC-AUC | ECE | C@80 |
| AmbigQA | L | 44.8 | 0.731 | 0.316 | 17.4 | 41.9 | 0.724 | 0.287 | 12.4 |
| | D | | 0.767 | **0.114** | 16.1 | | 0.763 | **0.085** | 13.0 |
| | R | | **0.821** | 0.120 | **26.2** | | **0.812** | 0.147 | **20.6** |
| SQA-Temp | L | 35.7 | 0.757 | 0.223 | 9.4 | 29.1 | 0.751 | 0.157 | **3.0** |
| | D | | 0.772 | **0.086** | 6.7 | | 0.757 | **0.048** | 2.7 |
| | R | | **0.797** | 0.119 | **13.5** | | **0.784** | 0.167 | 2.4 |
| SQA-Geo | L | 35.6 | 0.759 | 0.221 | 10.4 | 29.0 | 0.757 | 0.156 | 3.3 |
| | D | | 0.743 | **0.085** | 8.9 | | 0.723 | **0.056** | 4.5 |
| | R | | **0.800** | 0.120 | **13.7** | | **0.789** | 0.176 | **4.6** |

Table 2: Ambiguous Calibration. Column **M** refers to the calibration methods, abbreviated with (L)ikelihood, (D)iversity, and (R)epetition. Loose matching counts the answer as correct if it matches the answer from any interpretation; strict matching only counts it as correct if one of the closest interpretations has the same answer.

| Dataset | M | N | ROC-AUC | ECE | C@80 |
|---|---|---|---|---|---|
| NQ | D | 3 | 0.749 | 0.132 | 40.8 |
| | | 5 | 0.791 | 0.086 | 44.2 |
| | | 8 | 0.806 | 0.125 | 43.9 |
| | | 10 | 0.814 | 0.152 | 44.8 |
| | R | 3 | 0.789 | 0.155 | 43.3 |
| | | 5 | 0.811 | 0.116 | 45.0 |
| | | 8 | 0.826 | 0.099 | 45.3 |
| | | 10 | 0.830 | 0.103 | 45.4 |
| AmbigQA | D | 3 | 0.650 | 0.297 | 12.1 |
| | | 5 | 0.719 | 0.218 | 12.5 |
| | | 8 | 0.755 | 0.148 | 16.8 |
| | | 10 | 0.767 | 0.114 | 16.1 |
| | R | 3 | 0.769 | 0.152 | 16.7 |
| | | 5 | 0.801 | 0.130 | 24.3 |
| | | 8 | 0.815 | 0.120 | 26.9 |
| | | 10 | 0.821 | 0.120 | 26.2 |

Table 3: Calibration by number of samples (N). EM is excluded, because it is solely based on the greedy answer and does not depend on the number of samples.

| Setting | M | EM | ROC-AUC | ECE | C@80 |
|---|---|---|---|---|---|
| NQ | L | 65.8 | **0.833** | **0.098** | 70.7 |
| | D | | 0.807 | 0.142 | **74.7** |
| | R | | 0.797 | 0.137 | 71.2 |
| AmbigQA | L | 57.6 | 0.698 | 0.356 | 27.3 |
| | D | | 0.718 | 0.148 | 28.1 |
| | R | | **0.794** | **0.137** | **52.9** |

Table 4: Calibration for instruction tuned Flan-PaLM models. The column labeled **M** refers to the calibration methods, abbreviated with (L)ikelihood, (D)iversity, and (R)epetition.

any follow-up work investigate how the uncertainty in interpreting a question interacts with uncertainty about the answer.

## 8.1 Calibration and Selective QA

Calibration of modern machine learning approaches is difficult due to their non-linearity, scale, and architecture details (Guo et al., 2017). Generation-like tasks are even more difficult, containing a sequence of tokens with individual confidence estimates and a special end of sentence marker that may be poorly calibrated (Kuleshov and Liang, 2015; Kumar and Sarawagi, 2019; Jagannatha and Yu, 2020). A common approach for various natural language processing tasks is to train a separate classifier to estimate the model's confidence (Kumar and Sarawagi, 2019; Jiang et al., 2021; Zhang et al., 2021b; Kamath et al., 2020; Desai and Durrett, 2020; Dong et al., 2018) using various featurizations of the model's input, output, and likelihoods. Alternatively, Ren et al. (2023) use embedding distances and Osband et al. (2021) use Bayesian-inspired approaches that try to predict the full distribution over labels.

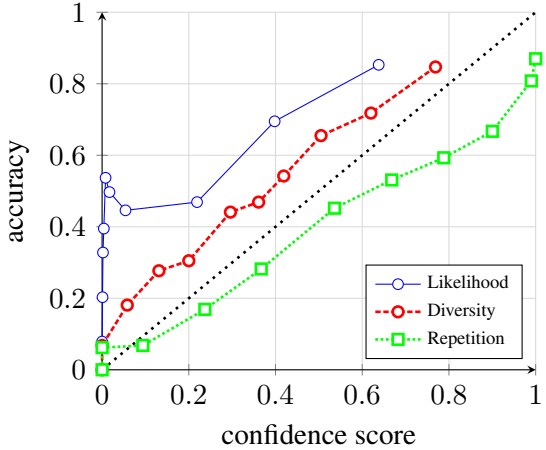

Figure 4: Plot of calibration error by comparing bucketed accuracy to bucketed confidence scores across methods. Plotted on the combined version of AmbigQA, containing ambiguous and unambiguous questions.

Within the specific setting of question answering, Kamath et al. (2020) and Zhang et al. (2021b) both address a similar selective question answering framework to ours, but they do not explore ambiguous questions. Varshney et al. (2022) investigate selective prediction for various tasks, and Varshney and Baral (2022) aim to improve coverage by re-answering questions that the model originally abstained on. Kuhn et al. (2023) and Lin et al. (2023) use similar sampling-based methods over free-form question answering, using slightly different formulations of confidence scores, but they do not investigate ambiguous questions. Kuhn et al. (2022) examine synthetically-created ambiguous questions, but focus on multi-turn interactions.

Question ambiguity can be formalized as uncertainty about the denotation of the question, relating to prior work on uncertainty in semantic parsing (Dong et al., 2018). However, rather the expressing candidate denotations in a formal semantic representation, we express them informally in natural language, as a waypoint toward quantifying the uncertainty of candidate answers.

### 8.2 Self-Calibrating Language Models

A number of papers propose to quantify confidence and detect hallucinations by (a) directly asking the model to rate its own confidence or correctness (e.g., Kadavath et al., 2022; Lin et al., 2022), (b) drawing repeated samples (Wang et al., 2022; Manakul et al., 2023), or (c) requesting additional supporting information about proposed outputs (Agrawal et al., 2023). As discussed in Section 5 and Appendix C, we found that neither self-verification nor offering the option of "unknown" were effective strategies for selective question answering. Wang et al. (2022) focus on accuracy rather than calibration and Manakul et al. (2023) focus on fact-checking long-form answers.

### 8.3 Prompting Strategies

In in-context learning, the choice of prompt and the order of exemplars can affect downstream performance (Lu et al., 2022). Other work suggests that giving more long-form answers and explanations in the prompt may encourage the model to do the same, increasing its likelihood of arriving at the correct answer (Wei et al., 2022). In our work, we find the choice of prompt plays little role and that longer answers or explanations do not seem to improve calibration (see Appendix B). Zhou et al. (2023) investigate how adding words of uncertainty interacts

with calibration, noting that additional expressions of uncertainty improve calibration without hurting accuracy. This is an interesting direction for future research into its interaction with ambiguity.

## 9 Conclusion

We investigate the calibration of large language models, extending prior work by distinguishing uncertainty about the answer (epistemic uncertainty) from uncertainty about the meaning of the question (denotational uncertainty). We propose a disambiguate-and-answer paradigm, where the model first attempts to rephrase the question before providing its answer. This paradigm enables straightforward techniques for quantifying model confidence by counting the frequency of answers within repeated samples from the language model. These sample-based confidence metrics are particularly effective for ambiguous questions, and are significantly better calibrated than traditional likelihood-based measures and self-verification. For ambiguous questions, the proposed method is similar to calibration through self-consistency.

## 10 Limitations

In this work, we explore selectively answering ambiguous questions and calibrating such models, but we do so only within the context of a single model: PaLM. We do not explore alternative paradigms, such as supervised fine-tuning, or other large language models for in-context learning with which to replicate our results. Furthermore, we only explore calibration in a closed-book setting, despite that open-book question answering (i.e., augmented with retrieval) generally has higher performance and may pose additional calibration challenges. Moreover, our exploration of various confidence scores was relatively brief, and while we explored some additional prompts, we did not iterate heavily on prompt tuning. Lastly, while the proposed method has some advantages, especially for instruction-tuned models on ambiguous questions, it also increases the compute linearly with the number of samples.

## Acknowledgements

We thank William W. Cohen, Urvashi Khandelwal and Clara Meister for their valuable comments and feedback while discussing this work.

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

## A  Ambiguity Prediction

A full description of how we transformed our method into ambiguity prediction is below.

**Disambiguate and Answer**  In the prompting setup described, the model should produce an interpretation of an ambiguous question before answering it. Thus, if the model produces an interpretation, it predicts that it is more like the ambiguous questions in the prompt than the unambiguous ones. We can thus use this as a prediction of ambiguity. We use both the greedy output (*Greedy Disambig.*) and the sampled output as a voting mechanism (*Voting Disambig.*).

**Disagreements and Unique Answers**  These methods are simply the inverse of Sampling Repetition and Sampling Diversity described in Section 3. *Num Disagreements* refers to the fraction of sampled answers that are not the same as the greedy answer; *Num Answers* refers to the fraction of unique answers produced in the sampling procedure. As previously discussed, a more diverse set of answers could reflect either question ambiguity (denotational uncertainty) or epistemic uncertainty.

**Direct Prediction**  We also experiment with a different few-shot prompt where instead of predicting an answer, the task is to predict either the string *Ambiguous* or *Unambiguous*. We can then use the greedy output as a binary label (*Greedy Direct*) or use sampling as a voting mechanism (*Voting Direct*).

As stated in the main text, none of the methods are very effective. An exploration of the precision/recall tradeoff can be found in Figure 5.

## B  Chain of Thought

Chain-of-thought prompts can improve performance by providing examples of the reasoning patterns that yield valid answers (Wei et al., 2022). As a simple chain-of-thought, we have the model

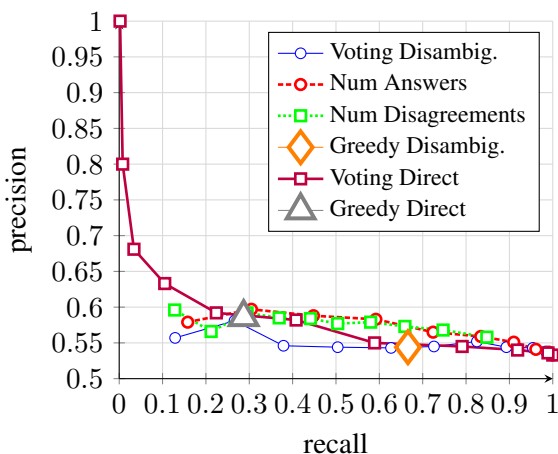

Figure 5: Precision vs recall for ambiguity prediction. For the sampling-based methods, each point corresponds to a classification threshold corresponding to counts over the ten sampled outputs. The greedy predictions are plotted as single points. None of these systems improve precision over the baseline rate of 53%.

first produce the long answer and then the short answer, or vice versa. These long answers are taken from ASQA for AmbigQA or the provided long answers for Natural Questions, and we refer to these conditions as 'Long + Short' and 'Short + Long', depending on whether the short answer is placed before or after the long answer. From Table 6, we can see this has small and mostly negative effects on Natural Questions: the strongest effect comes from the decline in accuracy when the long answer is produced first, because the model sometimes hits its max decode length. For ambiguous questions, found in Table 7, there is a small gain in accuracy at the cost of every calibration metric.

In the direction of chain of thought, does the choice of exemplars in the prompts matter? Here, we select the questions by hand and write clear long answers and use these as the exemplars in the prompt. This also did not have much impact on performance, so we only do this investigation for the unambiguous questions, and call it 'Static Short + Long', 'Static Long + Short', and 'Static Short Only' (for the original case but with the hand selected questions). We find that the prompt largely does not matter, achieving relatively similar results regardless of prompt, though many better prompts may exist. Results can be found in Table 6 for Natural Questions.

| | M | Natural Questions | | | | AmbigQA | | | |
|---|---|---|---|---|---|---|---|---|---|
| | | EM | ROC-AUC | ECE | C@80 | EM | ROC-AUC | ECE | C@80 |
| 540B | L | 51.7 | 0.843 | 0.141 | 44.2 | 44.8 | 0.731 | 0.316 | 17.4 |
| | D | | 0.814 | 0.152 | 44.8 | | 0.767 | 0.114 | 16.1 |
| | R | | 0.830 | 0.103 | 45.4 | | 0.821 | 0.120 | 26.2 |
| 62B | L | 38.8 | 0.823 | 0.111 | 18.8 | 34.6 | 0.690 | 0.235 | 3.4 |
| | D | | 0.826 | 0.203 | 24.2 | | 0.757 | 0.045 | 0.8 |
| | R | | 0.836 | 0.151 | 24.5 | | 0.805 | 0.150 | 3.4 |
| 8B | L | 16.0 | 0.829 | 0.038 | 0.8 | 14.8 | 0.699 | 0.121 | 0.6 |
| | D | | 0.813 | 0.228 | 0.8 | | 0.747 | 0.080 | 0.0 |
| | R | | 0.800 | 0.162 | 0.8 | | 0.814 | 0.148 | 0.4 |

Table 5: Scaling results on the unambiguous and mixed version of Natural Questions / AmbigQA. Note that the 540B model is the same model as reported earlier in Section 5 and Section 6.4, reshown here for comparison. We use L, D, and R to mean Likelihood, Diversity, and Repetition for brevity.

| Natural Questions | | | | | |
|---|---|---|---|---|---|
| Prompt | M | EM | ROC-AUC | ECE | C@80 |
| Static | L | 50.5 | 0.756 | 0.474 | 8.0 |
| Long | D | | 0.820 | 0.117 | 36.3 |
| + Short | R | | 0.838 | 0.084 | 42.8 |
| Static | L | 48.7 | 0.844 | 0.196 | 39.2 |
| Short | D | | 0.817 | 0.194 | 38.0 |
| + Long | R | | 0.824 | 0.177 | 37.0 |
| Long | L | 42.5 | 0.656 | 0.42 | 4.2 |
| + | D | | 0.793 | 0.147 | 22.7 |
| Short | R | | 0.848 | 0.079 | 31.0 |
| Short | L | 49.6 | 0.835 | 0.154 | 37.3 |
| + | D | | 0.820 | 0.168 | 38.0 |
| Long | R | | 0.832 | 0.133 | 42.5 |

Table 6: Experiments on Natural Questions using different prompt formats. In the first, we hand-wrote the prompt. In the second set, we randomly chose exemplars from Natural Questions. Note that the randomly selected exemplars are sometimes quite long, causing the model sometimes to run out of decoding space in that setting.

| AmbigQA | | | | | |
|---|---|---|---|---|---|
| Prompt | M | EM | ROC-AUC | ECE | C@80 |
| Long | L | 36.3 | 0.663 | 0.152 | 1.2 |
| + | D | | 0.758 | 0.041 | 6.0 |
| Short | R | | 0.859 | 0.061 | 21.8 |
| Short | L | 45.7 | 0.716 | 0.351 | 12.4 |
| + | D | | 0.742 | 0.127 | 13.8 |
| Long | R | | 0.795 | 0.123 | 21.8 |

Table 7: Experiments on AmbigQA using different prompt formats. We randomly chose exemplars from the ASQA annotations over Natural Questions. Note that the randomly selected exemplars are sometimes quite long, causing the model sometimes to run out of decoding space in that setting. Note that we do not hand-write prompts here due to the difficulty of writing intentionally ambiguous questions well.

## C   Text as Calibration

Can language models simply tell us when they don't know the answer? Here, we use a prompt where two of the questions are given the answer "Unknown", instead of the given answer. The method follows the above prompting strategy, except two of the answers are randomly replaced with Unknown, and the unambiguous and ambiguous both use six exemplars. Results can be found in Table 8. Overall, on the unambiguous portion of Natural Questions, the model chooses to abstain a relatively large fraction of the time for only slightly higher accuracy. The sampling repetition method is able to answer 86.1% of the questions at 80% accuracy, which is substantially better.

The ambiguous results are more interesting, where the model very rarely outputs unknown even though the questions are supposedly more challenging. The model is able to improve its accuracy somewhat by not answering a small number of questions; however, it cannot reliably abstain when it is wrong, thus leading to an overall low percentage of accuracy. While not computed above, the standard model's C@45 would be approximately 99.6%.

| Method | Acc | Cov |
|---|---|---|
| **Natural Questions** | | |
| Long + Short | 49.8 | 65.3 |
| Short + Long | 62.1 | 65.4 |
| Short Only | 60.4 | 73.0 |
| Static L+S | 61.9 | 53.7 |
| Static S+L | 63.9 | 53.7 |
| Static SO | 63.9 | 63.7 |
| **AmbigQA** | | |
| Long + Short | 38.8 | 86.4 |
| Short + Long | 46.6 | 91.3 |
| Short Only | 45.7 | 95.2 |

Table 8: Answer or Unknown Results. Note that the Acc is the EM rate only on the answered questions, so they are most comparable to the Cov@Acc numbers. While they are somewaht difficult to compare to the above numbers as neither Cov or Acc is fixed, this method is generally uniformly worse than the other methods.

## D  Scaling

See Table 5 for results. Note that calibration as defined by error is roughly constant, the proportion of the questions that can be answered with reasonable accuracy declines dramatically, approximating zero on the smallest model sizes.

## E  Sampling

See **??** for results.