# OpenReview forum: "Selectively Answering Ambiguous Questions"
_EMNLP/2023/Conference — EMNLP 2023 Main_

### Official Review · Reviewer_DXG6 · 2023-08-03

**Soundness:** 4

**Excitement:**

4: Strong: This paper deepens the understanding of some phenomenon or lowers the barriers to an existing research direction.

**Paper Topic And Main Contributions:**

This paper breaks down the uncertainty in answering a (potentially ambiguous) question into the uncertainty associated with understanding the question and the uncertainty in giving the right answer due to a lack of knowledge.  The authors then present multiple candidate formulations of model confidence and evaluate these when asked to answer ambiguous and unambiguous questions in a few-shot manner. The candidates presented by the authors are based on sampling diversity and repetition which are calculated on a set of 10 generated outputs for each question. These confidence measures are compared to two from existing work - using the model likelihood and using model self-verification via prompting.

The sampling repetition confidence score obtains good performance on a suite of calibration metrics across both kinds of questions. While model likelihood is a strong candidate on unambiguous questions, its performance drops considerably on ambiguous ones highlighting the connection to the breakdown of uncertainty. Finally, the authors also present experiments that show that the sampling-based confidence scores do not degrade when fine-tuning the model whereas likelihood gets worse.

**Questions For The Authors:**

1. Since the sampling-based results can yield different outputs on multiple runs, can we obtain some statistics like standard deviations on these results?


2. Why was the self-verification baseline not included in Table 2?


3. For section 7, were any of these datasets used for experiments directly used in the FLAN Palm fine-tuning? This could be a confounder in the improvement in performance/calibration.

**Reasons To Accept:**

The paper is written and structured very well making it easy to follow from the general breakdown of uncertainty motivating the experiments and results.


The results show clear results of trends across various calibration-based metrics that the sampling-based confidence scores, while simple and intuitive, are more effective than the two baseline methods.


The main paper is self-contained with the experiments to evaluate the basic research questions and the appendix has a number of interesting related results that I had as questions, including scaling trends and variations in prompt format.

**Reasons To Reject:**

Hard to state it as a 'weakness' per se, but given that the paper proposes a general formulation it would be very helpful to see if these trends follow in other (open-source) models as well. The paper reports multiple variants and do list this as a limitation as well but I think it's worth mentioning since many readers won't be able to access the Palm model.

**Reproducibility:**

1: Could not reproduce the results here no matter how hard they tried.

**Reviewer Confidence:**

1: Not my area, or paper was hard for me to understand. My evaluation is just an educated guess.

---

> ### Author Rebuttal · Authors · 2023-08-29
>
> Thanks for your careful review.
>
> ## Re: Standard deviation of performance
> This is a very interesting request. Unfortunately, computing even a relatively inaccurate standard deviation based on three samples would require running all of our experimental conditions multiple times which is computationally expensive and not practical in the limited time available for the author response. We ran our methods twice on Natural Questions, and were somewhat satisfied that nothing shifted directionally and there were only small deviations.
>
> ## Re: Self-verification baseline
> We did not run self-verification on the ambiguous questions because it performed poorly on the unambiguous questions and seemed like an unnecessary use of resources. Beyond that, the method to convert the prompt to handle multiple interpretations of a question was somewhat unclear. However, if the reviewer thinks this experiment would improve the final version, then we can run it.
>
> ## Re: NQ in the instruction tuned set
> Neither Natural Questions nor AmbigQA are used to train FLAN. TriviaQA was used in the FLAN fine-tuning data, which is one reason we excluded this dataset from section 7.

---

### Official Review · Reviewer_x5aR · 2023-08-04

**Typos Grammar Style And Presentation Improvements:** 1.	The paper oscillates between absta…
**Soundness:** 3

**Excitement:**

2: Mediocre: This paper makes marginal contributions (vs non-contemporaneous work), so I would rather not see it in the conference.

**Missing References:**

N/A

**Paper Topic And Main Contributions:**

The paper discusses uncertainty in question answering setup from two perspectives—the uncertainty in the meaning of the question (denotational uncertainty) and uncertainty in the answer (epistemic uncertainty). The paper argues that the existing work in confidence calibration is only done with unambiguous questions and extends this to ambiguous questions leading to denotational uncertainty. The paper proposes two resampling based confidence estimation strategy (1) # times the sampled output matches the greedy output and (2) the ratio of number of unique samples over total samples). The paper argues that these confidence estimates capture uncertainty more accurately both for ambiguous and unambiguous questions.

**Questions For The Authors:**

1.	The paper mentions that for ambiguous questions, “the prompt will first include an interpretation of the question which is one of the provided disambiguations” (lines 412-414). What is the idea behind doing this? If the prompt already includes the disambiguation of the question, is the question even ambiguous anymore?
2.	The paper discusses that the sampling diversity method is “heavily dependent on the number of samples, as we do not expect the number of unique answers to scale linearly with the number of samples” (line 217). Is there a way to normalize for the number of samples to avoid this effect? How would this affect selecting a threshold for practical purposes?
3.	The paper mentions “This leads us to our disambiguate-then-answer paradigm: if there is no fundamental difference between ambiguous and unambiguous queries when taken out of the original context, …” (lines 453-456). What does this even mean? Ambiguous queries will be inherently ambiguous taken out of context, as opposed to the unambiguous queries. Can you please clarify what the point here is?
4.	For the self-verification baseline, the model is first prompted to provide a list of possible answers to a given question, followed by a second task to determine if each answer is correct or incorrect.  The paper mentions that the candidate set of answer may contain duplicates. Wouldn’t this affect the evaluation? Wouldn’t repetition lead to probabilities being split between multiple answer? Why not consider the unique set of candidate answer before calculating confidence?

**Reasons To Accept:**

1.	The problem of studying uncertainty calibration in the case of ambiguous questions is quite interesting.  However, I think the proposed experiments are severely lacking.

**Reasons To Reject:**

1.	The main contributions of the paper seem exaggerated. The paper claims its first main contribution as “we reframe the discussion of model confidence with respect to denotational and epistemic uncertainty”. Lin et al. (2023)---one of the papers cited in this submission---already brings out this distinction as “uncertainty, which depends only on the input, and confidence, which additionally depends on the generated response.” The paper includes no discussion or reference to this overlap in the key idea. The second main contribution of the paper is “we present two simple and effective approaches for measuring confidence under all types of uncertainties”. Again, the paper later reveals that the resampling based confidence estimates are previously also proposed in Kuhn et al. (2023) and Lin et al. (2023). I feel that this misrepresents the contribution of the paper initially, where eventually the only emerging distinguishing factor from existing literature is additional experiment on ambiguous question.
2.	The experimental setup and evaluation design does not align well with the motivated problem. For evaluating calibration, the paper considers the confidence calibration metrics that are standard in the literature, such as Expected Calibration Error, ROC-AUC, etc. However, for this evaluation, the paper “consider(s) the answer as correct if it matches the answer of any of the disambiguation” (line 463-464). This defeats the whole purpose of model being able to abstain when there is denotational uncertainty in the question if the model is rewarded for answer to any possible interpretation of the question. What is the objective of this evaluation and how is this evaluation serving its intended purpose?
3.	The paper mentions various works in uncertainty calibration literature, such as Zhang et al. (2021b), Varshney et al. (2022), and Lin et al. (2023), which propose host of different methods beyond likelihood or self-verification for confidence calibration. However, the paper does not include most or all of the proposed methods in these works as baselines. Based on the discussion of the prior literature in the paper, it is not clear if the baselines discussed in the paper are the best possible baselines and good representative of the different approaches in the existing literature.
4.	The likelihood-based baseline and resampling based methods vary in compute cost as the latter require multiple queries before processing a single query (to decide whether to answer or abstain). How does that factor into the differences in the baseline and the “proposed method”. Further, how do confidence estimates depend on the number of samples and what is a good number of samples to be able to obtain reliable confidence estimates? These are some key discussions that are missing from the paper that are important to address.

**Reproducibility:**

3: Could reproduce the results with some difficulty. The settings of parameters are underspecified or subjectively determined; the training/evaluation data are not widely available.

**Reviewer Confidence:**

4: Quite sure. I tried to check the important points carefully. It's unlikely, though conceivable, that I missed something that should affect my ratings.

---

> ### Author Rebuttal · Authors · 2023-08-29
>
> Thank you for your careful review. As the reviewer mentions, the focus of our paper is indeed calibration over ambiguous questions, which we tried to make clear in the title and abstract. However, the reviewer mentions that discussions of the various types of uncertainty are present in other work. However, do note that the focus on ambiguous questions meant that denotational uncertainty is more likely to be present. We can edit the summary of contributions to re-emphasize that handling ambiguous questions was the main contribution. Please also note that the work mentioned is extremely contemporaneous with our work, with Lin et al (2023) being released after the EMNLP anonymity period.
>
> For answering the ambiguous questions, the prompt's exemplars contained demonstrations of disambiguating the question before answering it, which the model was expected to do as well. For the question the model was to answer, it did not see the interpretation of the question, it instead had to produce it.
>
> Previously, we only gave the model credit if it's produced interpretation was closest to the interpretation matching the answer -- however, this made little difference in practice, and there were many ways to measure if the interpretation matched the answer. In general, those results lowered the accuracy but kept calibration roughly the same:
>
>
> --|Likelihood | Diversity | Repetition
> -----|-----|-----|-----
> | ROC/AUC | 0.724 | 0.762 | 0.814  |
> | ECE |        0.284 | 0.082 | 0.150 |
> | Cov@80 |     12.4 | 13 | 21.2 |
>
> Comparing to the original numbers, nothing is directionally different. We thought this was an aside and not worth centering in the discussion, but can try to clarify this with the additional space provided.
>
> ## Re: Number of Samples
> On selecting for the number of samples, there are some suggestions (e.g., Menhinick's index suggests dividing by the square root), but probably using a dev set to choose the correct value will always be useful in practice. While time/resource limitations prevent us from checking more samples for now, we did check 3, 5, and 8 (by subsampling from our original 10) for NQ (Table 1) and AmbigQA (Table 2). For the Sample Repetition confidence metric, the ROC-AUC and C@80 decreased by less than 2% when the number of samples was reduced from 10 to 5. For the Sample Diversity confidence metric, the impact of reducing the number of samples was somewhat larger, especially on the AmbigQA dataset. Both metrics were considerably worse when using only 3 samples. Indeed, as the reviewer notes, more samples increases the cost linearly, so it is important to get the number correct.
>
> For NQ:
>
> |–|n|roc_auc|Cov@80|ece|
> |----|----|----|----|----|
> |D|3|0.749|40.8|0.132|
> |D|5|0.791|44.2|0.086|
> |D|8|0.806|43.9|0.125|
> |D|10|0.814|44.8|0.152|
> —
> |R|3|0.789|43.3|0.155|
> |R|5|0.811|45.0|0.116|
> |R|8|0.826|45.3|0.099|
> |R|10|0.830|45.4|0.103|
>
>
> For AmbigQA:
> |–|n|roc_auc|Cov@80|ece|
> |----|----|----|----|----|
> |D|3|0.650|12.1|0.297|
> |D|5|0.719|12.5|0.218|
> |D|8|0.755|16.8|0.148|
> |D|10|0.767|16.1|0.114|
> —
> |R|3|0.769|16.7|0.152|
> |R|5|0.801|24.3|0.130|
> |R|8|0.815|26.9|0.120|
> |R|10|0.821|26.2|0.120|
>
>
> ## Re: " Ambiguous queries will be inherently ambiguous taken out of context, as opposed to the unambiguous queries."
> The goal of the method was not to abstain when there was denotational uncertainty in the question. In fact, our position was that there is always some amount of denotational uncertainty in the question (usually by accident). Instead, we wanted the model to only answer if it was confident it 'knew what the user meant'. We will try to clarify the wording of this in the final version. The point is that among natural questions, many queries are unintentionally ambiguous because of a lack of shared context, and queries that seemingly have only one interpretation to one reader may still depend on time and place in interesting ways. Thus, ambiguity is a function of the reader (or model, in this case), not just the question, and re-stating an interpretation of the question is one way to determine if the reader/model understood correctly. This is part of our argument for why a model might disambiguate and answer instead of abstain-when-ambiguous.
>
> ## Re: Self-verification duplicates
> The model is a decoder-only model and the answer spans do not compete with each other for probability. Instead, the model sees the list of candidate answers as well as the proposed greedy answer (similar to the prompt described in the cited paper). It then scores the token TRUE. Thus, the only competition for probability is between TRUE and all of the other tokens in the vocabulary, and nothing in the prompt is scored at all. We will clarify this method in the final version.
>
> ## Re: Other baselines
> Many previous baselines look at multiple choice question answering, reading comprehension question answering reframed as multiple choice question answering  (Zhang et al 2021b)., or sentence classification (Varshney et al, 2022). There are methods in e.g., Lin et al (2023) could potentially be used; however, they unfortunately do not release numbers that we can compare against directly. As that paper was contemporaneous with us, we were unable to re-implement their methods for this submission.

---

### Official Review · Reviewer_RjGt · 2023-08-05

**Soundness:** 4

**Excitement:**

4: Strong: This paper deepens the understanding of some phenomenon or lowers the barriers to an existing research direction.

**Paper Topic And Main Contributions:**

This paper tackles the task of calibration in QA, specifically considering ambiguous questions that may have multiple interpretations. The contribution of the paper are as follows:
- it frames and analyzes the calibration of ambiguous questions (AQ) in terms of two different categories based on the source of uncertainty: denotational and epistemic uncertainty.
- it presents two approaches for measuring confidence for both types of uncertainty that is better than the simple likelihood-based calibration and self-verification prompt of previous work.

**Questions For The Authors:**

A. Results show that sample repetition method improves calibration significantly from likelihood method for ambiguous questions, but results for self-verification method is missing from Table 2. How does the self-verification method hold for calibration of ambiguous questions?

**Reasons To Accept:**

- While works that consider ambiguous questions in QA and calibration exist, few work study them together and this paper presents a novel insight into the difficulty of calibration of AQ in terms of the two categories of AQ.
- The proposed sampling repetition calibration method is simple to implement and well motivated.

**Reasons To Reject:**

- The repetiton sampling mechanism itself is similar to self-consistency decoding method [1] , and thus limited in novelty. However, its application to calibration in this paper still seems novel and reasonable.

[1] Wang et al., Self-Consistency Improves Chain of Thought Reasoning in Language Models, ICLR 2023.

**Reproducibility:**

4: Could mostly reproduce the results, but there may be some variation because of sample variance or minor variations in their interpretation of the protocol or method.

**Reviewer Confidence:**

3: Pretty sure, but there's a chance I missed something. Although I have a good feel for this area in general, I did not carefully check the paper's details, e.g., the math, experimental design, or novelty.

---

> ### Author Rebuttal · Authors · 2023-08-29
>
> Thank you for your careful review. As mentioned, this can be thought of as a form of self-consistency; however, unlike self-consistency, we are using the greedy (temperature=0) answer rather than the majority-vote answer and focus on ambiguous questions and calibration.
>
> We did not run self-verification on the ambiguous questions because it performed poorly on the unambiguous questions and seemed like an unnecessary use of resources. Beyond that, it is not completely clear how to convert the self-verification prompt to handle multiple question interpretations. However, if the reviewer thinks this experiment would improve the final version, then we can run it.

---

### Meta-Review · Area_Chair_51ty · 2023-09-11

**Recommendation:** 5

**Metareview:**

This paper studies calibration of question answering systems when presented ambiguous questions. It first decomposes uncertainty in this setting into uncertainty about interpreting the question vs. uncertainty in the right answer (due to lack of knowledge). The paper also proposes different ways to measure model confidence; the best method samples 10 outputs and measures their similarity. The reviewers were generally supportive of this paper, finding that it proposed an important setting and a method that performs well. The paper also includes valuable analysis of how things like model scale, prompt format, and fine-tuning affect results.

One reviewer was concerned about the evaluation setting used, as the paper currently considers a model to be correct if it outputs any correct answer to any valid interpretation of an ambiguous question, even if that interpretation differs from the model's predicted interpretation. During the rebuttal, the authors presented additional results where the model is only considered correct if its answer matches its chosen interpretation. These results show the same patterns as before. I agree with the reviewer who raised this concern that this issue merits further discussion in the camera-ready version, and that both sets of results should be presented (either in the main text or appendix, depending on space considerations).

---

### Decision · Program_Chairs · 2023-10-07

**Decision:**

Accept-Main

**Comment:**

This paper studies calibration of question answering systems when presented ambiguous questions. It first decomposes uncertainty in this setting into uncertainty about interpreting the question vs. uncertainty in the right answer (due to lack of knowledge). The paper also proposes different ways to measure model confidence; the best method samples 10 outputs and measures their similarity. The reviewers were generally supportive of this paper, finding that it proposed an important setting and a method that performs well. The paper also includes valuable analysis of how things like model scale, prompt format, and fine-tuning affect results.

One reviewer was concerned about the evaluation setting used, as the paper currently considers a model to be correct if it outputs any correct answer to any valid interpretation of an ambiguous question, even if that interpretation differs from the model's predicted interpretation. During the rebuttal, the authors presented additional results where the model is only considered correct if its answer matches its chosen interpretation. These results show the same patterns as before. I agree with the reviewer who raised this concern that this issue merits further discussion in the camera-ready version, and that both sets of results should be presented (either in the main text or appendix, depending on space considerations).